# Exosome-Mediated Cellular Communication in the Tumor Microenvironment Imparts Drug Resistance in Breast Cancer

**DOI:** 10.3390/cancers17071167

**Published:** 2025-03-30

**Authors:** RamaRao Malla, Priyamvada Bhamidipati, Anuveda Sree Samudrala, Yerusha Nuthalapati, Vasudevaraju Padmaraju, Aditya Malhotra, Annah S. Rolig, Sanjay V. Malhotra

**Affiliations:** 1Cancer Biology Group, Cancer Biology Laboratory, Department of Life Sciences, GITAM School of Science, GITAM (Deemed to be University), Visakhapatnam 530045, Andhra Pradesh, India; 2Center for Experimental Therapeutics, Knight Cancer Institute, Oregon Health & Science University, Portland, OR 97201, USA

**Keywords:** breast cancer, drug resistance, drug sensitization, exosomes, tumor microenvironment

## Abstract

**:** Breast cancer (BC), a leading cause of cancer death in women, exhibits heterogeneity and drug resistance, hindering treatment efficacy. This review examines how exosomes, key mediators of tumor microenvironment communication, contribute to BC drug resistance. We delve into exosomal mechanisms conferring resistance to established chemotherapies and immunotherapies. The review further explores existing systematic approaches in elucidating exosome-drug resistance relationship and highlights promising therapeutic strategies to overcome exosome-mediated resistance in BC.

## 1. Introduction

Breast cancer (BC) is the most diagnosed cancer among women and the leading cause of cancer-related death. Worryingly, BC incidence is increasing in adult women aged <50 years worldwide. This increase in risk of early BC appearance may be due to exposure to carcinogens in early and young stages of life or may be the result of lifestyle, diet, and environmental risk factors, which induce alterations in the microbiome and the genome and cause genetic susceptibilities [1].

While BC is the leading cause of cancer-related mortality for women, prevalence rates and mortality rates vary among different countries and different BC subtypes. BC is more prevalent in Asian and African countries, and BC prevalence rates are rapidly increasing in Asian countries like China and South Korea, while they are declining in the USA [2]. Furthermore, there is reduced mortality in developed nations compared to underdeveloped and developing nations. The treatment of BC is highly challenging due to its heterogeneity. Additionally, BC mortality varies across BC subtypes, which include luminal A and B (estrogen receptor (ER) and progesterone receptor (PR) positive), HER2-positive, basal-like, and triple-negative breast cancer (TNBC). Patients with TNBC have higher mortality rates in comparison to other BC subtypes, in part due to high incidence of brain, bone, liver, and lung metastasis [3]. For luminal A and B type BCs, which are hormone receptor-positive, hormone therapy, like hormone receptor inhibitors (e.g., tamoxifen [TMX]), stands out as one of the most efficient treatment approaches. Targeted therapies, like HER2 antibodies (e.g., trastuzumab [TRA]), have shown promise in treating HER2-positive BC [4]. Unfortunately, limited targeted therapies exist for TNBC [5], and thus chemotherapy (e.g., paclitaxel [PTX], doxorubicin [DOX], and cisplatin [CP]) is the standard-of-care therapy for TNBC due a lack of specific targets. However, PARP inhibitors for TNBC with germline BRCA mutation and immune checkpoint inhibitors (ICIs) for advanced TNBC with high programmed death-ligand 1 (PD-L1) expression have recently been approved by the FDA [6]. Two important barriers to treating highly fatal subtypes of BC, particularly TNBC, are the rapid advancement to metastasis and a lack of early symptoms, which makes diagnosis difficult.

Drug resistance significantly impacts survival rates in BC subtypes, and drug resistance to therapeutic strategies is a key cause of high mortality. The usage of anticancer treatments can be limited by either primary (de novo) or acquired treatment resistance. While de novo resistance is primarily related to tumor features that exist before the use of anticancer medicines, for example, cancer cell adhesion to stromal components, including the extracellular matrix (ECM) [7], in contrast, acquired resistance mechanisms develop in response to exposure to treatments [8]. The mechanism(s) of drug resistance can be internal or external, stemming from genetic and phenotypic alterations within the tumor or from the surrounding milieu, respectively. Indeed, TNBC cells utilize many methods to evade cell death caused by chemotherapeutic agents. These methods can involve pumping the drug out of the cell, deactivating it [9], initiating bypass signaling or pro-survival processes, increasing the repair of damaged DNA, and fostering the epithelial-mesenchymal transition (EMT) and stem-like characteristics [10,11,12].

Understanding the key mechanisms that underlie treatment resistance is central for developing more potent therapies that reduce metastasis and improve patient outcomes. This review examines the multifaceted nature of exosomes in mediating treatment resistance within the complex tumor microenvironment (TME) of BC. We thoroughly examine the latest progress in comprehending how exosomes contribute to resistance against well-known chemotherapeutic drugs like TMX, PTX, DOX, and CP, along with ICIs. Furthermore, we explore promising therapeutic strategies to overcome exosome-driven drug resistance in BC, highlighting potential avenues for improved treatment efficacy.

## 2. Tumor Microenvironment

The TME is a complex network, made up of both cellular and non-cellular components, like stromal cells, including cancer-associated fibroblasts (CAFs) and cancer-associated adipocytes (CAAs), along with non-stromal cells, including cancer cells, cancer stem cells (CSCs), endothelial cells (ECs), and immune cells, including T and B lymphocytes, natural killer cells (NKs), and tumor-associated macrophages (TAMs), in addition to blood and lymphatic networks, ECM, pH, and partial pressure of oxygen (pO2). The TME is a vital component of tumorigenesis, tumor progression, and response to treatment [13], and it performs key roles in the development of drug resistance in part by facilitating communication between stromal cells, non-stromal cells, and immune cells [13] through direct cell-to-cell interactions, signaling molecules and exosomes. In addition, non-cellular components of TME also contribute to tumor progression; the ECM acts as physical barriers to drugs [14], while acidic pH limits the uptake of the drug [15], and hypoxia modulates various genetic alterations [16]. Low pH [17,18] and hypoxia [19,20] promote the exosomes’ release and uptake, facilitate cellular communication, and potentially contribute to metastasis. These studies highlight the significant contribution of exosomes in the development of drug resistance under low pH and hypoxia.

CSCs with high self-renewal capacity and tumorigenic ability contribute to treatment resistance. CAA promote drug resistance by communicating with CSCs in the TME [21]. TAMs are the most prominent immune cells in the BC TME, which promote treatment resistance by polarizing cancer cells into CSCs [22]. Recent studies have found that exosomes from TAMs promote tumor progression and dissemination by transferring proteins, non-coding RNAs, and metabolites to tumor cells within the TME [23]. EC-derived exosomes can also promote progression of cancer by inducing angiogenesis via transporting pro-angiogenic biomolecules and suppressing expression of factors that inhibit hypoxia-inducible factor 1 (HIF-1) [24].

### 2.1. Exosome Biogenesis and Secretion in the TME

Exosomes, nanosized extracellular vesicles with diverse bioactive chemicals, are crucial mediators of cellular communication. The biogenesis, packing, and release of exosomes from donor cells like tumor cells, breast cancer stem cells (BCSCs), and stromal cells like CAFs and their uptake by recipient cells like neighboring cancer cells, T cells, endothelial cells, and macrophages has been comprehensively reviewed [25]. The mechanisms of exosome biogenesis, secretion, and internalization in TME involve many components [26]. These components include ESCRT, which is made of multimeric protein complexes (ESCRT-0, -I, -II, and -III) and other proteins that sort endosomal proteins and macromolecules into intraluminal vesicles [27], and lipid rafts and tetraspanins, which sort microRNAs (miRs) and long non-coding RNAs (lncRNAs) into exosomes.

The ESCRT pathway is a fundamental cellular machinery involved in multivesicular body (MVB) formation and cargo sorting for exosome biogenesis [28]. This pathway is utilized by both cancer cells and TME cells [29,30,31]. Cancer cells often dysregulate the ESCRT pathway to promote exosome secretion for immune suppression, metastasis, and drug resistance [32]. Oncogenic tetraspanin, like CD151, targets exosomes to the lung and lymph nodes [33] and promotes tumor progression [34]. Additionally, Rab, a protein from a family of small GTPases, aids in the sorting of cargo and secretion of exosomes [35].

### 2.2. Mechanisms of Exosome-Mediated Intercellular Communication in TME

Exosomes are key communicators within the TME, mediating local as well as systemic cell-to-cell communication (Figure 1). Exosomes communicate by horizontal transmission of DNA, mRNA, miRs, and proteins, among others [36]. Through this transmission, they function as signal transducers between cancer cells and other cells in the TME [37,38] and reprogram the TME to promote tumors to develop drug resistance and become metastatic [39]. Furthermore, exosomes produced in the TME enhance carcinogenesis and drug resistance by activating essential oncogenic pathways in cancer cells [40]. These mechanisms exist constitutively, but drug presence intensifies them.

Tumor cell-derived exosomes (TDEs) promote the formation of the BC premetastatic niche (PMN) and BC metastasis to lungs by transferring caveolin-1. They induce the expression of genes associated with the BC premetastatic niche in lung epithelial cells and triggers the secretion of tenascin-C, which in turn instigates ECM deposition and facilitates macrophage polarization to M2-type macrophages. M2 macrophages stimulate angiogenesis in lung tissue by secreting VEGF-A [41]. At the premetastatic niche, TDEs reprogram tumor cell metabolism by transferring miR-122, which facilitates metastasis by increasing nutrient availability [42]. TDEs trigger M2 macrophage activity by stimulating NF-κB-dependent secretion of IL-6, which induces a TLR-2-mediated pro-inflammatory phenotype in the TME and promotes drug resistance.

In cancer immunotherapy, cytotoxic T cells (TC cells) mainly execute the elimination of cancer cells. However, mutations in the apoptotic pathway led to drug resistance due to failure of Ag-specific TC cell-mediated elimination of cancer cells [43]. TDEs induce drug resistance by reprogramming TC cells. To do so, exosomes transfer cirmiR-20a-5p, which directly reduces nuclear protein ataxia-telangiectasia (NPAT) expression by interacting with its 3′-UTR regions. This reduction in NPAT expression contributes to TC cell dysfunction by enhancing PD-1 expression [44]. TDEs also induce TC cell dysfunction by downregulating the AKT-mTOR pathway to reduce TC cell glycolysis [45]. Finally, TDEs induce an immunosuppressive TME phenotype by reducing CD3+HLA-DR+ T cells, increasing CD3+PD-L1+ T cells, and promoting IL-10 secretion from CD4+CD127-CD25hi Tregs [46].

The TDEs from drug-resistant BC cells can induce drug resistance by transferring their cargo and mechanistically triggering EMT, remodeling the TME, promoting a stem cell-like phenotype, modulating drug efflux mechanisms, and activating survival signals [47]. TDEs also reprogram normal fibroblasts into CAFs in the BC TME by transferring SNHG14, which positively regulates FAM171A1 expression via EBF1 [48].

CAFs are the most abundant cells in the BC TME, and they promote BC progression through exosome secretion [49]. CAF exosomes induce BC cell proliferation and metastasis by transferring miR-500a-5p, which directly targets the tumor suppressor gene *USP26* by interacting with its UTR region. This promotes metastasis by downregulating E-cadherin expression and upregulating N-cadherin, FN1, ZEB1, Snail, and Slug [50]. CAF exosomes promote drug resistance in BC cells by transferring mutant mitochondrial DNA, which abolishes oxidative phosphorylation. Consequently, quiescent cancer cells exit dormancy by reprogramming their metabolism and increasing self-renewal [51]. CAF exosomes promote tumor cell invasion by transferring autophagy-associated GPR64, which stimulates non-canonical NF-κB signaling to upregulate MMP9 and IL-8 in recipient BC cells, thus enabling cancer cell invasion [52].

TDEs promote angiogenesis by transferring EPHA2 to endothelial cells. Ephrin A1 ligand-dependent EPHA2 induces AMPK signaling, which induces VEGF expression via activation of HIF-1α-dependent transcription [53]. In the TME, BC cells use bidirectional communication to activate CAAs. CAA-derived exosomes promote drug resistance and metastasis by transferring circCRIM1. circCRIM1 inhibits miR-503-5p, thus activating glycosylation hydrolase, which decreases FBP1 protein stability. Furthermore, both glycoside hydrolase and FBP1 promote immune cell infiltration [54]. These studies highlight the exosome-mediated crosstalk between cells in the TME, which transfers cargo that promotes drug resistance.

## 3. Exosomes Impart Drug Resistance in BC

BCs are primarily characterized by their ability to invade and metastasize. These processes involve complex interactions between cancer cells and adjacent cells in TME at each stage. As intercellular communicators, exosomes can directly transfer their cargo into cells, activating oncogenic signaling pathways that promote invasion and metastasis [55]. Additionally, exosomes play a crucial role in the development of drug resistance in BC [56]. For example, exosomes are essential for transferring functional resistance proteins among cancer cells, which can regulate processes that enhance chemotherapeutic drug efflux. The exosomes also transfer molecules imparting drug resistance either unidirectionally or bidirectionally between cancerous and non-cancerous cells within the TME [57,58]. Researchers recognize that exosomes facilitate at least three mechanisms that enhance the efflux of chemotherapeutic agents: they directly facilitate drug efflux, they modulate the expression of proteins and miRNAs that govern specific membrane-associated drug efflux pumps, including P-glycoprotein (P-gp) [59], and they reduce intracellular drug concentrations through packing and transporting drugs out of the cells [60].

### 3.1. Exosome-Mediated Tamoxifen Resistance in BC

Tamoxifen (TMX) is a targeted therapeutic used for treatment of ER-positive BC. The mechanism of action of TMX is dependent on the presence of the ER. It binds to the ER, blocking estrogen from binding and inhibiting the growth-promoting effects of estrogen on BC cells [61,62]. However, drug resistance in BC patients, acquired through various exosome-driven mechanisms, can lead to the termination of TMX treatment (Figure 2). For instance, exosomes from drug-resistant cancer cells impart drug resistance by delivering the circular RNA (circRNA) circ_UBE2D2 to drug-sensitive cancer cells, which upregulate ER-α expression and promote metastasis through interaction with miR-2001-3p [63]. This study found that circ_UBE2D2 interacts with miR-2001-3p, an anti-oncogene, and inhibits its activity. This inhibition leads to upregulation of ER-α expression. While robust expression would traditionally predict TMX sensitivity, this sensitivity is paradoxically bypassed by alternative mechanisms such as PI3K/AKT or MAPK pathways, conferring resistance. In addition, the upregulation of ER-α expression enhances metastasis by promoting EMT, cell migration, and invasion. Another study demonstrated that TMX-resistant MCF-7-cell-derived exosomes transfer miR-9-5p, which reduces drug-induced apoptosis in parental MCF-7 cells by negatively regulating adiponectin expression, a gene that controls metastatic properties such as adhesion, invasion, and migration [64]. Xu et al. demonstrated that exosomes released from TMX-resistant MCF-7 cells, LCC2, are enriched with urothelial carcinoma-associated 1 (UCA1) and transfer TMX tolerance to drug-sensitive MCF-7 cells. These UCA1-enriched exosomes induce TMX resistance by reducing caspase-3 expression in recipient cells and thus inhibiting apoptosis [65]. Furthermore, inhibition of apoptosis is a key mechanism by which cancer cells evade treatment and develop drug resistance by upregulating antiapoptotic proteins and survival signaling pathways [66].

A bioinformatics study identified genes associated with TMX resistance. Heat shock protein family H (Hsp110) member 1 (HSPH1), a key gene associated with TMX resistance, is enriched in exosomes of MCF-7 cells [67]. Zhao et al. showed that exosomes from TMX-resistant MCF-7 cells can transfer TMX resistance to wild-type MCF-7 cells by transferring miR-205 to suppress drug-induced apoptosis. miR-205 inhibits caspase-3 by binding to the transcription factor E2F1 and inhibiting AKT phosphorylation [68]. The study by Gao et al. demonstrated that SFRS1 mediates the packing of miR-22 into exosomes of cells in the TME, such as CAFs, which induce TMX resistance in recipient breast cancer cells by the binding of miR-205 to ER*α* and PTEN. In addition, exosomes contribute to the maintenance of CAF phenotypes by transferring CD63, which activates STAT3 signaling [69]. These studies describe exosome-mediated TMX resistance mechanisms in BC.

### 3.2. Exosome-Mediated Doxorubicin Resistance in BC

The anthracycline DOX is a chemotherapeutic drug used to treat TNBC (Figure 3). The mechanisms of DOX anticancer activity are DNA intercalation, topoisomerase II inhibition, and free radical production [70]. Increased drug efflux and changes in topoisomerase II expression are two mechanisms that contribute to resistance against DOX [71].

In a recent mechanistic study, DOX-resistant BC cell lines showed increased levels of exosomal miR-181b-5p. When exo-miR-181b-5p was added, they actively fused with drug-sensitive MCF-7 cells to convey a drug-resistant feature. Overexpression of miR-181b-5p conveys drug resistance by suppressing BCLAF1 expression, reducing DOX-induced G1 arrest and senescence, and lowering p53/p21 levels. Additionally, an exosomal-miR-181b-5p inhibitor resulted in tumor management, reversal of the DOX-resistance phenotype, and increased tumoral BCLAF1 expression [72].

Recently, Zhang et al. showed that exosomes induced drug tolerance in HER2-positive BC by transferring oncogenic circHIPK. Functional studies showed that exosomes derived from drug-resistant cells can transfer drug resistance to recipient cells. The exosome-mediated drug resistance mechanism was conferred through circHIPK3 overexpression, which serves as a competing endogenous RNA (ceRNA) that directly associates with miR-582-3p. miR-582-3p regulates RNF11expression, a protein involved in protein ubiquitination, by direct interaction at complementary sites in the seed region. The circHIPK/miR-582-3p-driven RNF11 induces drug resistance by selective degradation of proapoptotic proteins [73].

The studies by Santos et al. found that exosomes obtained from breast cancer stem cells (BCSCs) are resistant to DOX and PTX. Mechanistically, BCSC-derived exosomes containing miR-155 trigger drug resistance in drug-sensitive recipient cells. In these cells, miR-155 triggers EMT and generates DOX-resistant CSCs [74]. Additional studies revealed that exosomes derived from MCF-7 BC cells imparted DOX resistance. One study showed that miR-222 drives DOX resistance in BC cells by activating the PTEN/AKT/FOXO signaling pathways. Indeed, suppressing the PTEN/AKT/FOXO1 axis sensitizes DOX-resistant cells by inducing apoptosis [75].

One study linking miR-222 to DOX resistance in BC revealed that resistance was mediated by a pathway that reduced p27kip1 synthesis. By directly targeting CCNG2, exosome transfer of miR-1246 to BC cells induced a tumor-promoting phenotype that included drug resistance [75]. In addition, research has identified a close relationship between Notch signaling and BC development and recurrence. Exosomes transport and transfer miR-34a-5p, which effects DOX tolerance by lowering the intracellular concentration of the protein through modulating miR-34a-5p/Notch1 [76]. Recently, exosome-delivered lncRNA-GAS5 was shown to reduce ABCB1-driven DOX efflux by suppressing miR-221-3p, which targets Dickopf Wnt signaling pathway inhibitor 2 (DKK2) and consequently triggers the Wnt/β-catenin signaling axis, which reduces ABCB1 expression [77].

Hong et al. described how exosomal miR-221-3p mediates DOX-resistant BC cells. They showed that DOX-treated cells secrete exosomes enriched with miR-221-3p. Further, the exosomes transfer drug resistance by delivering miR-221-3p, which targets phosphoinositide-3-kinase regulatory subunit 1 (PI3KR1), a tumor suppressor and a key negative regulator of PI3K activity [78,79]. Inhibition of PIK3R1 by mirR221-3p may lead to DOX resistance. These studies emphasize the importance of exosomes in transmitting DOX resistance in BC.

### 3.3. Exosome-Mediated PTX Resistance in BC

Paclitaxel (PTX) is a taxane derivative frequently employed as a chemotherapeutic agent for TNBC (Figure 4). The mechanisms of PTX activity encompass various pathways by which PTX influences cellular functions, leading to programmed cell death [80]. For example, PTX stabilizes tubulin to stabilize microtubule composition. This prevents the dynamic instability of microtubules required for spindle assembly and cell division during mitosis. This ultimately leads to cell death without affecting DNA or RNA synthesis [81]. PTX is widely known for its ability to inhibit the cell cycle at the G2/M phase and trigger apoptosis. This is mediated by disrupting normal microtubule polymerization and depolymerization by attaching to tubulin. Further, PTX leads to cell death by activating cleavage of procaspases as well as pro-apoptotic markers. Therapeutic studies indicate that after prolonged PTX chemotherapy, over 50% of patients exhibited significant PTX resistance and tumor progression. The tolerance of TNBC to PTX is a significant challenge in clinical practice, as PTX resistance is a primary contributor to mortality linked to treatment failure. Consequently, drug resistance constitutes the primary obstacle to PTX therapy in TNBC [82,83].

Factors influencing PTX resistance include ATP binding cassette (ABC) transporters, miRs, and genetic alterations in specific genes. Proteins associated with PTX drug efflux include P-gly, MRP-1, and BC resistance protein (BCRP). This drug efflux leads to a decrease in drug efficacy and an increase in drug resistance [84]. Additionally, BC cells may induce changes to their metabolic mechanisms that bring about drug resistance. BC cells can develop PTX resistance through changing protein expression of efflux pumps, molecular alterations to the apoptotic pathway, and an upregulation of PTX resistance-associated gene 3 (TRAG-3/CSAG2). In addition, PTX makes the cancer cells resistant to chemotherapy through upregulation of multidrug resistance genes, changes in efflux pump activity, and alterations in apoptotic pathways, thus evading chemotherapeutic effect on tumor cells [85].

Santos et al. studied how miR-155 overexpression is related to EMT in PTX-resistance BC cells. In these studies, when PTX-sensitive cells and CSCs received exosomes from PTX-resistant cells, miR-155 concentration increased along with PTX resistance. Furthermore, transfection with miR-155 activated EMT [74]. Xia et al. showed that exosomes derived from PTX-treated BC cells made recipient cells more PTX resistant by turning on the circBACH1/miR-217/G3BP2 signaling pathway. Upon binding to cirBACH1, miR-217 targets GTPase-activated SH3 domain-binding protein2 (G3BP2), resulting in G3BP2 overexpression, which mediates metastasis and promotes PTX resistance [86]. The EZH2/STAT3 pathway is activated in BC cells, which results in the release of PTX-induced miR-378a-3p and miR-378d rich exosomes. Upon internalization, these exosomes activate the WNT and Notch stem cell cycles, which govern cellular growth and differentiation and promote cancer cell stemness and other tumor cell characteristics attributed to drug resistance [87,88,89]. Furthermore, the STAT3 pathway activates drug resistance by promoting ZEN to directly attach to miR-378a-3p and miR-378d promoter regions, which increases their expression in the exosomes [87].

The study on MCF-7 BC stem cell-derived exosomes containing ANXA6 revealed that the release of exosomes containing ANXA6 boosts PTX resistance in BC by activating autophagy. Exosomal ANXA6 activates autophagy by stimulating YAP1 to abnormally increase the Hippo pathway, which promotes PTX resistance [90]. Apart from ANXA6, lncRNAs are also involved in regulating drug resistance in BC.

PTX-resistant BC cell-derived exosomes overexpress survivin, another protein that facilitates evasion of apoptosis and transfer of PTX resistance. A study by Kreger et al. found that PTX treatment induces exocytosis of survivin-enriched exosomes from MDA-MB-231 cells. These survivin-rich exosomes are transferred to neighboring cells, increasing the probability of survival in the neighboring recipient cells by suppressing apoptosis, thus conferring resistance to PTX treatment. Furthermore, there is evidence shown that plenty of lncRNAs, when delivered by exosomes during treatment, ensure the viability of chemoresistant BC cells [91]. Exosome-meditated transfer of HSP90 induces degradation of p53 in PTX-sensitive cells, thereby increasing PTX resistance [92]. These studies highlight the mechanistic role of exosomes in conferring PTX resistance in BC.

### 3.4. Exosome-Mediated Platinum Drug Resistance in BC

Platinum-based drugs, such as cisplatin (CP), carboplatin (CB), oxaliplatin (OXA), nedaplatin, and lobaplatin, are therapeutically effective against various cancers. In fact, the most common drugs used to treat BC and its highly metastatic subtype, TNBC, are CP, CB, and OXA. They exert their cytotoxic effects by forming DNA adducts, primarily intra-strand crosslinks. These adducts distort DNA structure, inhibiting replication and transcription, which triggers cell cycle arrest and ultimately apoptosis in cancer cells [93].

Exosomes can mediate platinum-based drug resistance through various mechanisms, including bypassing apoptosis, inducing autophagy, changing the TME, and promoting angiogenesis and immune evasion (Figure 5). Furthermore, exosomes aid the outflow of the drug from the cell, thus reducing the concentration of drug in the cancer cells, thereby diminishing cytotoxic effects. Exosomes also promote drug resistance by altering tumor cell metabolism. Recent experimentation indicates that exosomes also play a part in drug resistance by delivering miRNAs, lncRNAs, and various other proteins that promote cell survival pathways and reduce drug accumulation in cancer cells [55].

CP inhibits DNA replication and cell proliferation. Exosomes upregulate the ATP-binding cassette transporter, P-gp, thus increasing export of CP from cancer cells and inducing drug resistance, migration, and invasion while reducing apoptosis. The investigation done by Wang et al. shows that exosomes isolated from CP-resistant MDA-MB-231 cells alter the sensitivity of wild-type MDA-MB-231 to CP, which is mediated by exosomal miR-423-5p [94].

Research by Cheng et al. elucidated a novel mechanism by which Snail-regulated exosomal miR-21 enhances CP resistance in cancer cells. They demonstrated that Snail-induced EMT in tumor cells leads to an increase in miR-21-rich exosomes. miR-21 inhibits PTEN and BRCC3, accelerating phosphorylation and ubiquitination of NLRP3, which prevents inflammasome assembly. Thus, these exosomes suppress NLRP3 inflammasome activity in TAMs, thereby reducing the inflammatory response to chemotherapy. This potent interplay between cancer cells and the TME highlights the importance of targeting non-genetic adaptation mechanisms to overcome chemotherapeutic resistance [95].

Superoxide dismutase 1 (SOD1) is an enzyme that is highly active in TNBC stromal cells. This enzyme promotes cancer progression and metastasis and is difficult to counteract. The exosomes derived from SOD1 fibroblasts contain miR-3960. This miR-3960 enhances CP resistance by targeting BRSK2 activity. When delivered to a tumor cell, miR-3960 downregulates BRSK2 expression, which lowers S16 phosphorylation of PIMREG, enhancing its stability and initiating the NF-κB signaling pathway [96]. Research has shown that the EMT also influences the progression of resistance to platinum-based therapies. Therefore, targeting the EMT process, identifying exosome biomarkers, and understanding the exosomal pathways involved in platinum drug resistance could provide an avenue to overcoming platinum drug resistance in BC.

### 3.5. Exosome-Mediated Trastuzumab Resistance in BC

Trastuzumab (TRA) is a targeted therapeutic used for treatment of HER2-positive BC (Figure 6). TRA targets the HER2 protein in HER2 overexpressing cancers. It selectively binds to the HER2 receptor, disrupting signaling pathways that promote cancer cell growth and survival [4,97]. Despite this, long-time treatment triggers TRA resistance, leading to reduced efficacy and overall survival of patients. Liu et al. found that overexpression of Linc00969, a non-coding RNA, in exosomes derived from TRA-resistant patients, and transferring TDEs containing Linc00969 induce a TRA-resistant phenotype in HER2-positive BC cells that were formally sensitive to TRA. Further, the influence of exosomal transfer of Linc00969 on the development TRA resistance was confirmed using both silencing and overexpression of Linc00969 in functional studies on BC cells. These studies showed that Linc00969 induces upregulation of the HER-2 gene, and the interplay with HUR maintains the stability of HER-2 mRNA. In addition, exosomal Linc00969 promotes TRA resistance by triggering autophagy [98]. Mechanistically, Linc00969 promotes overexpression of the HER-2 gene by interacting with HUR, an RNA-binding protein. This activates compensatory pathways like PI3K/AT or MAPK pathways and bypasses the drug’s inhibitory effect. This interaction also saturates TRA binding efficacy, thus leading to resistance.

Zhang et al. reported that exosomes stimulate drug resistance in HER2-positive BC by transmitting the oncogenic ci-rRNA circHIPK. They established this exosome-mediated drug resistance mechanism using rescue assays and found that overexpressed circHIPK3 acts as a ceRNA. circHIPK3 is regulated by miR-582-3p, which regulates RNF11 expression, a protein involved in ubiquitination, by direct interaction at a complementary site in the seed region. This RNF11 driven by circHIPK/miR-582-3p induces drug resistance by selectively degrading proapoptotic proteins [99]. Martinez et al. demonstrated that exosomes from TRA-resistant BC cells that contain the neuropeptide neuromedin U (NmU) induce drug resistance and immune evasion. Mechanistically, exosomes enriched with NmU enhance the release of TGF-β and PD-L1 from cells. TGF-β induces TRA resistance, and PD-L1 induces immune suppression in BC [100]. These studies reveal the distinctive capacity of exosomes to transfer TRA resistance to TRA-sensitive BC cells. Exosomes drive drug resistance in different subtypes of BC. Exosomal contents/major pathways involved/mechanisms driving drug resistance in BC subtypes are summarized in Table 1.

### 3.6. Exosome-Mediated ICI Resistance in BC

Immunotherapy is a promising and pivotal tool to treat cancers due to its targeted approach and compatibility with combinational therapies. This approach is particularly relevant for TNBC and HER2+ BC, the aggressive forms of BC, because of their high expression of programmed cell death ligand 1 (PD-L1) and the presence of tumor-infiltrating lymphocytes (TILs) [106].

Immune checkpoint inhibitors (ICIs) assist the host’s immune system in directly attacking cancer cells. ICIs enhance immune system eradication of tumor cells by blocking immune checkpoint proteins like PD-L1, programmed cell death protein 1 (PD-1), and cytotoxic T-lymphocyte-associated antigen-4 (CTLA-4), which act as brakes on immune cells. Among all the ICIs approved by the FDA (U.S. Food and Drug Administration), most of them are anti-PD-1 (pembrolizumab and dostarlimab), anti-PD-L1 (atezolizumab, durvalumab), or anti-CTLA-4 drugs (ipilimumab). For BC patients, these agents can be used either as monotherapy or in combinational therapy [106,107].

Exosomes from BC cells have been found to contain PD-L1, CTLA-4, and T-cell immunoglobulin and mucin domain 3 (TIM-3) [108]. The presence of these checkpoint receptors in exosomes can trigger resistance to immunotherapy and help in cancer progression. PD-L1 is enriched in many exosomes [109] and adheres to PD-1 receptors on T-cells, triggering the PD-1 signaling pathway in which SH protein tyrosine phosphatase 2 (SHP2) is activated. This decreases the PI3K/AKT axis, thus suppressing T-cell activity. The approved drugs that target PD-1 are pembrolizumab, nivolumab, cemiplimab, dostarlimab, and retifanlimab, and those that target PD-L1 are avelumab, durvalumab, and atezolizumab [110]. Unfortunately, tumor exosomes carrying PD-L1 can inhibit T cells even when PD-1 inhibitors are bound to PD-1 on T cells, thus effectively bypassing the immune checkpoint blockade and allowing the cells to evade immune surveillance [111]. This resistance, however, can be sensitized using anti-PD-L1 antibodies to restore T cell activity.

CTLA-4 is an immune checkpoint protein found on T cells, including constitutive expression on regulatory T (Treg) cells. Tumor exosomes carrying CTLA-4 suppress T cell activity and promote tumor growth. CTLA-4 binds with high affinity to the B7 ligands on antigen-presenting cells (APCs), and thus CTLA-4 binds competitively with CD28, a surface protein on T cells that helps activate and sustain them. The approved drugs for CTLA-4 inhibition are ipilimumab and tremelimumab, which are often used in combination with pembrolizumab and other chemotherapies.

Recent studies have revealed that exosomes carrying transmembrane proteins like galectin-9 can bring T cell death. Other proteins found on exosomes that induce T cell inhibition are immunoglobulin, TIM-3, ARG1, T-cell immunoreceptor-based inhibition motif domain (TIGIT), lymphocyte activation gene 3 (LAG-3), B and T lymphocyte attenuator (BTLA), and V-domain immunoglobulin suppressor of T-cell activation (VISTA) [112].

Furthermore, exosomes obtained from metastatic BC cells can overpower T cell activity and block NK cell cytotoxic and immune surveillance activity. Exosomes can alter the TME by passing on immune suppressive molecules like TGF-β and IL-10 to suppress immune cell function [47]. Exosomes secreted from mesenchymal stem cells can metamorphose myeloid cells into immunosuppressive macrophages, which promote BC; although the exact mechanisms are unclear, hypotheses include contributions of the transcription factor HIF1α and a hypoxic TME [113]. Exosomes from BC cells can also alter the TME by secreting exosomal miR-222, which directly targets PTEN, activates the AKT cascade, and induces macrophage polarization towards an immunosuppressive M2 phenotype [75].

Exosomes carrying miRNAs and lncRNAs alter the gene expression, making the tumor cell resistant to ICIs. Exosomal miR-222 is obtained from BC cells by targeting the PTEN/AKT pathway [75].

Studies on coculture of tumor-derived exosomes and CD25− CD4+ T cells show that exosomes promote the production and proliferation of Tregs by transforming CD25− CD4+ T cells into FoxP3+ CD4+ CD25high T cells via microRNA-214-dependent suppression of PTEN in T cells and induction of Tregs to secrete IL-10, which ultimately leads to the promotion of tumor growth [114]. BCSC-derived exosomes induce T cells to generate Tregs by transferring FOXP3 [115]. Generally, mechanisms that promote resistance against ICIs enable cancer cells to escape immune surveillance. Immunosuppression and ICI resistance remain significant challenges to maximizing immunotherapy efficacy.

## 4. Targeting Exosome Formation and Secretion in BC

Exosome secretion is a critical step in exosome-mediated drug resistance. Therefore, inhibiting exosome secretion using a combination of clinically approved anticancer drugs or antagonists of key mediators could sensitize drug-resistant BCs to therapy (Table 2).

### 4.1. Inhibition of Exosome Secretion

Macitentan (MAC), a drug used clinically for pulmonary arterial hypertension, inhibits type 1 endothelial receptors (ETA). A recent study demonstrated that MAC inhibits secretion of exosomes containing PD-L1. Mechanistically, MAC synergizes with anti-PD-L1 to induce cell death in BC cells. Combining MAC with anti-PD-L1 reduced the interaction between T cell PD-1 and exosomal PD-L1 by blocking ETA. This combination reduced TNBC tumor growth in nude mice by increasing the number of cytotoxic T cells and reducing the number of Tregs in the tumor [116]. Another study reported that sulfisoxazole (SFX), an oral antibiotic, suppressed exosome secretion by reducing the expression of genes involved in exosome biogenesis and secretion by blocking ETA and promoted exosome degradation by enhancing exosome fusion with lysosomes in BC [117].

### 4.2. Sensitization of Exosome-Mediated DR in BC

A plethora of studies have documented that exosomes mediate drug resistance through diverse mechanisms. Studies have reported that targeting these exosome-mediated resistance mechanisms can sensitize drug-resistant cancer cells. A study by Kong et al. demonstrated that treating MDA-MB-231 BC cells with a combination of guggulsterone (GS), an antagonist of farnesoid X receptor (FXR), and bexarotene (BXT), an agonist of retinoid X receptor (RXR), reduced BCRP expression in cells by increasing the secretion of exosomes containing BCRP by elevating ceramide, which induces exosome secretion. The combination also reduced drug resistance by enhancing the retention of DOX and triggering apoptosis [118].

Chen et al. reported that d-Rhamnose β-hederin (DRβ-H) sensitized the docetaxel (DXL)-resistant MCF-7. Mechanistically, DRβ-H reduced the secretion of exosomes containing miR-16 from DXL-resistant MCF-7 cells and thus reduced the transmission of drug resistance to neighboring cells [119]. The combination of GW4869, a pharmacological inhibitor, and Nexinhib20, an inhibitor of Rab27, reduced the formation and release of exosomes. Nexinhib20 inhibits the exosome release by interacting with Tyr122, a critical residue of Rab27A pockets, via pi-pi stacking interactions. GW4869 blocks exosome release from multivesicular bodies (MVBs) by impeding the ceramide-dependent inward budding of MVBs via non-competitive inhibition of the membrane-neutral sphingomyelinase (nSMase2) [120].

**Table 2 cancers-17-01167-t002:** Inhibitors of exosome secretion and sensitizers of exosome-mediated drug resistance in breast cancer.

	Drug/Antagonist Name	Target	Mechanism/Mode of Action	Ref.
Inhibition of exosome secretion	Macitentan (MAC)	Type 1 ETA PD-L1	Synergizes with anti-PD-L1 Ab. Decreases PD-1/PD-L1 interaction. Blocks ETA. Increases the CTL count. Lowers the Treg count.	[116]
	Sulfisoxazole (SFX)	Biogenesis of exosomes type 1 ETA	Checks exosomal secretion by inhibiting the activity of genes involved in exosome biogenesis. Blocks ETA and controls the release of exosomes. Enhances the degeneration of exosomes through fusion with lysosomes.	[117]
Sensitization of exosome mediated drug resistance in BC	Guggulsterone + Bexarotene (GS + BXT)	FXR and RXR	Reduces BRCP expression in cells and the release of BRCP+ exosomes. Decreases drug resistance to DOX. Enhances apoptosis.	[118]
	d-Rhamnose-β-hederin (DRβ-H)	miR-16 containing exosomes	Sensitizes DXL-resistant MCF-7 cells. Limits transmission of drug resistance.	[119]
	GW4869	nSMase	Blocks the release of exosomes from MVBs.	[120]
	Nexinhib 20	Rab27	Reduces the formation and release of exosomes.	[120]

## 5. Discussion

Exosomes have recently become a central focus of cancer research due to their multifaceted roles as therapeutic targets and drug delivery agents.

Dong et al. presented a systematic approach toward understanding the relationship between exosomes and drug resistance in BC. They deliberated on attributes of exosome synthesis and changes in exosome cargo delivered by BC cells under drug intervention [47]. However, the studies are limited to in vitro models and small patient cohorts; hence, the clinical relevance of these findings remain unclear. Norouzi-Barough et al. explored the capacity of exosomes to predict the potency of chemotherapies and explain drug resistance in cancer cells [121]. These findings underscore the need to understand how exosomes can modify drug responses in women diagnosed with BC. Similarly, Dong et al. discussed the links between exosomes and treatment resistance in BC, emphasizing the uniqueness of exosome synthesis and the shift in exosome content in response to treatment [47].

Santos et al. pointed out the importance of exosomal miRNAs in effecting drug resistance and promoting the survival of neoplastic cells in the TME [122]. Antibodies against the surface of exosomes or compounds that inhibit exosome biogenesis could be used as agents that capture a previously unsuspected weak link in tumor cells, namely, the ability to metastasize and develop a drug-resistant phenotype. By focusing on the exosome signaling networks in BC, this brings an alternative approach to counter the disease’s drug resistance. In addition to treatment resistance, exosomes have been known to enhance BC tumor formation and progression [40], suggesting further reasons to target exosomes therapeutically. Thus, practically demonstrating the effective exosome- targeting strategies can bridge the scientific gap in clinical practice and contribute to improving the therapeutic outcomes.

Fontana et al. documented the exosomes and other extracellular vesicles that convey signals across tumor cells to adjacent stromal or immune cells in the TME, promoting treatment evasion and drug resistance. This clearly demonstrates that the exosome has multiple functions with respect to regulating the TME and influencing the process of cancer progression [123]. However, while the study highlights the versatility of exosomes, it lacks a systematic evaluation of which exosomal components play a key role in conferring resistance mechanisms. Furthermore, To et al. created an intervention map that highlighted the relationship between TME, exosomes, chemotherapy, and immunotherapy and stressed hypoxia-associated drug resistance. The topics most discussed were related to the presentation of exosomal signaling pathways under hypoxic conditions, which may trigger angiogenesis, invasion, metastasis, and immune evasion to promote treatment resistance in BC [124]. This underscores the importance of assessing the TME and the mode of exosome communication when designing rational treatment interventions for BC patients. Studies have shown that exosome-mediated cellular communication within the TME plays an important role in imparting drug resistance in BC. Understanding how exosomes influence drug responses and tumor progression can provide valuable insights for developing novel therapeutic approaches to overcome treatment resistance in BC patients [125]. Further work is required to determine the exact signaling cascades and molecular interactions underlying exosome secretion that cause drug resistance and to identify potential targets for therapeutic application in BC.

## 6. Conclusions

Drug resistance poses a major obstacle in the clinical management of BC. Exosomes serve as intercellular communication agents inside the TME, significantly influencing BC growth. A broader understanding of tumor heterogeneity has led to significant interest in the role of exosomes in the development of treatment resistance. Focusing on communication mediated by exosomes, specifically the transfer of resistance-inducing substances, constitutes a new path for therapeutic intervention. Further, deciphering the role of exosomes in transferring drug resistance from drug-resistant to drug-sensitive cells will enhance our current understanding. Researchers are trying to enhance the sensitivity of tumors to conventional treatments by disrupting exosome-mediated pathways. Studying distinct exosome functions and exosomal payload mechanisms offers the potential for developing innovative techniques for overcoming treatment resistance.

## Figures and Tables

**Figure 1 cancers-17-01167-f001:**
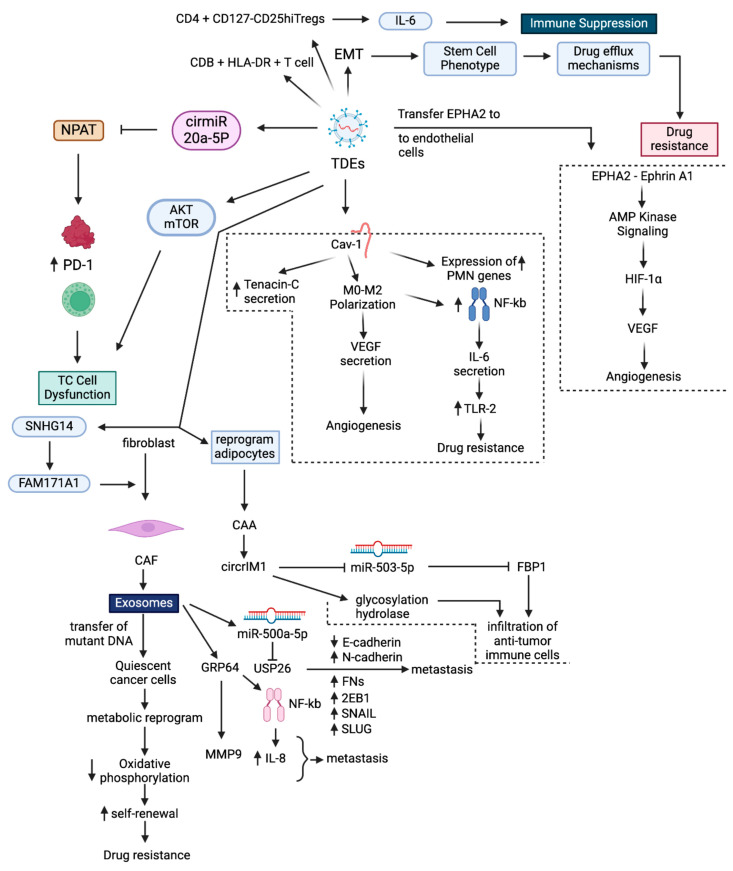
Interplay of exosomes and cellular communication in the BC TME. Tumor cell-derived exosomes (TDEs) initiate pre-metastatic niche formation in the lungs by transferring caveolin-1. This transfer induces the expression of premetastatic niche-associated genes in lung epithelial cells, stimulates tenascin-C secretion, and promotes extracellular matrix (ECM) deposition. Concurrently, TDEs facilitate macrophage (MØ) polarization to M2-type macrophages, which secrete VEGF-A to promote angiogenesis in lung tissues. Additionally, TDEs reprogram tumor cell metabolism by transferring miR-122, increasing nutrient availability to support metastasis. TDEs promote a pro-tumoral M2-like phenotype by activating NF-κB signaling, which leads to IL-6 secretion, which in turn activates toll-like receptor 2 (TLR2)-mediated pro-inflammatory response within the TME. TDEs suppress the anti-tumor immune response by reprogramming cytotoxic T cells (TC cells) via transfer of cirmiR-20a-5p. This transfer results in reduced expression of nuclear protein ataxia-telangiectasia (NPAT), which causes TC cell dysfunction by enhancing PD-1 expression and reducing glycolysis via disrupting AKT-mTOR signaling. TDEs also suppress T cells via reducing CD3+HLA-DR+ T cells, increasing CD3+PD-L1 T cells, and promoting IL-10 secretion from CD4+CD127-CD25hi Tregs. Drug-resistant cancer cells induce drug resistance phenotypes in sensitive cells by transferring exosomes that induce epithelial-to-mesenchymal transition (EMT), activate stem cell-like programs, modulate drug efflux mechanisms, and activate survival signals. TDEs facilitate angiogenesis by transferring ephrin type-A receptor 2 (EPHA2) to endothelial cells, thereby increasing vesicular endothelial growth factor (VEGF) expression through AMP-activated protein kinase (AMPK) signaling and HIF-1α activation. Cancer-associated adipocytes (CAAs) induce drug resistance and metastasis by transferring circular RNA cysteine-rich transmembrane bone morphogenetic protein regulator 1 (circCRIM1). This results in inhibition of miR-503-5p, which in turn activates glycosylation hydrolase, which destabilizes fructose-1,6-bisphosphatase 1 (FBP1). Both changes further promote immune cell infiltration. TDEs reprogram normal fibroblasts into cancer-associated fibroblasts (CAFs) by transferring small nucleolar RNA host gene 14 (SNHG14), which upregulates family with sequence similarity 171, member A1 (FAM171A1) expression through early B-cell factor 1 (EBF1). CAFs promote tumor progression by releasing exosomes. These exosomes transfer miR-500a-5p, which directly targets ubiquitin-specific peptidase 26 (USP26), leading to downregulation of E-cadherin and upregulation of N-cadherin, fibronectin 1 (FN1), zinc-finger E-box binding homeobox 1 (ZEB1), Snail, and Slug. Exosomes from CAFs transfer mutant mitochondrial DNA, which triggers quiescent cancer cells to exit dormancy, reprogramming their metabolism to abolish oxidative phosphorylation and increase self-renewal. CAFs also promote invasion by transferring autophagy-associated G-protein-coupled receptor 64 (GPR64), which upregulates matrix metalloproteinase 9 (MMP9) and interleukin-8 (IL-8) in recipient BC cells by stimulating non-canonical NF-κB signaling.

**Figure 2 cancers-17-01167-f002:**
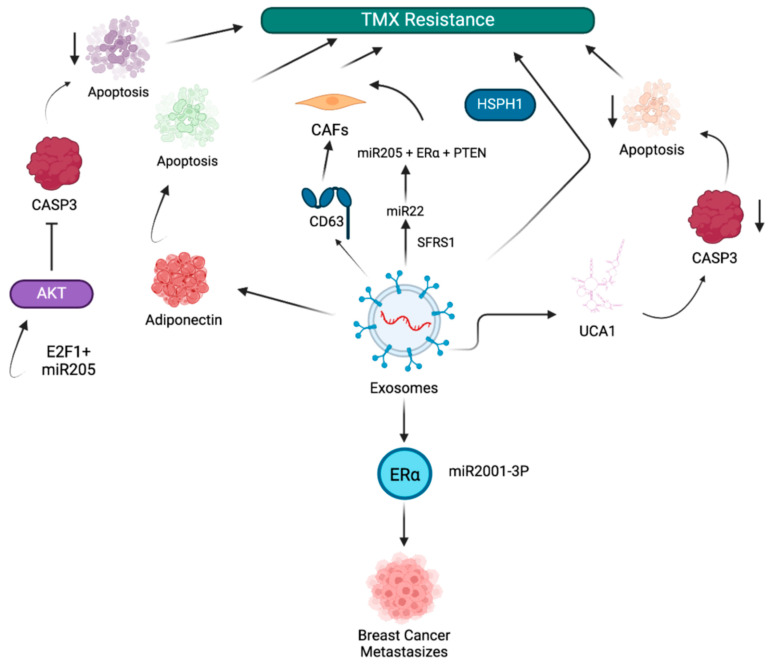
Exosome-mediated tamoxifen resistance in BC. Exosomes impart tamoxifen (TMX) resistance in several ways. By circular RNA encoding ubiquitin-conjugating enzyme E2 D2 (circUBE2D2) transfer, they promote BC metastasis. By mi-R-9-5p, miR205 transfer, and urothelial carcinoma associated 1 (UCA1) enriched exosomes, TMX resistance is induced by inhibiting apoptosis. Exosomes also maintain the cancer-associated fibroblast (CAF) phenotype by transferring CD63 via signal transducer and activator of transcription 3 (STAT3) signaling and by serine/arginine-rich splicing factor 1 (SFRS1) through miR-22 packing into them. Exosomal heat shock protein family H (Hsp110) member 1 (HSPH1) induces TMX resistance in MCF-7 cells. ETF1/miR205 induces TMX resistance by activating AKT, which inhibits caspase-3, reduces apoptosis, and enhances drug resistance.

**Figure 3 cancers-17-01167-f003:**
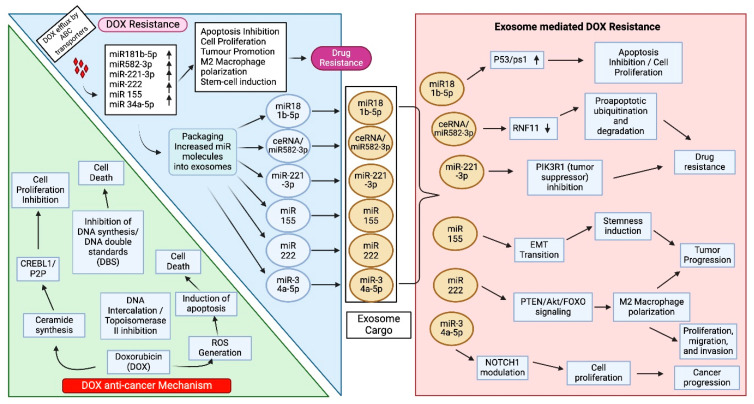
Exosome-mediated DOX resistance in BC. DOX induces toxicity in cancer cells by inhibiting cell proliferation, inducing DNA double-strand breaks (DBS), and inhibiting topoisomerase II-induced cell death by inducing ROS generation. Cancer cells develop resistance against DOX by increasing drug efflux through overexpressing ABC transporters and overexpressing miR181b-5p, miR582-3p, miR-221-3p, miR-222, miR 155, and miR 34a-5p, which promote apoptosis inhibition, cell proliferation, tumor promotion, M2 macrophage polarization, and stem cell induction. These miRs are lodged into exosomes and transported to DOX sensitive cells to induce resistance by multiple signaling mechanisms, as depicted in the figure. DOX inhibits cell proliferation by activating the cAMP responsive element binding protein-like 1 (CREBL1)/proliferation potential protein (P2P) pathway via enhancing synthesis of ceramide. Ring finger-11 (RFN-11) promotes ubiquitination and degradation of proapoptotic proteins; phosphatase and tensin homolog deleted on chromosome 10 (PTEN-10)/forkhead box O (FOXO) activates M2 macrophage polarization.

**Figure 4 cancers-17-01167-f004:**
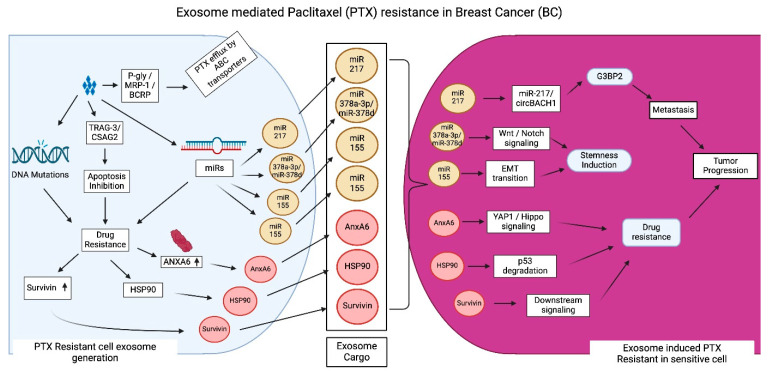
Exosome-mediated paclitaxel (PTX) resistance in BC: PTX induces toxicity in cancer cells by inhibiting tubulin to disrupt microtubule stability and affecting the process of cell division. It also activates procaspases and proapoptotic proteins, leading to cell death. Cancer cells develop resistance against PTX, increasing drug efflux through overexpressing ABC transporters (p-glycoprotein, multidrug resistance protein 1 and breast cancer resistance protein) and overexpressing miR217, miR378a-3p, miR378d, miR 155, annexin A6 (Anx6), heat shock protein 90 (HSP90), and survivin, which promote drug efflux, metastasis, EMT transition, stem cell induction, and tumor progression. These miR molecules and Anx6, HSP90, and survivin are lodged into exosome cargo and transported to PTX-sensitive cells to transfer resistance via multiple signaling mechanisms, as depicted in the figure. Exosomes mediate drug resistance by inhabiting apoptosis by activating Taxol resistance–associated gene-3 (TRAG-3)/chondrosarcoma-associated gene 2 (CSAG2).

**Figure 5 cancers-17-01167-f005:**
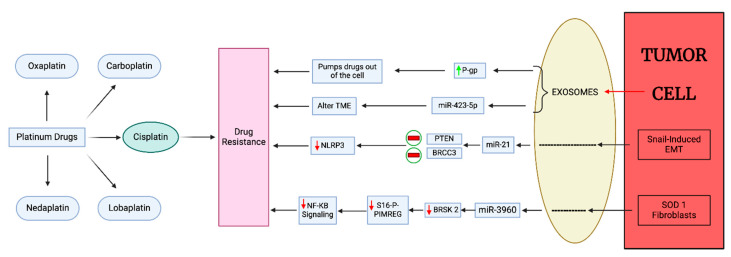
Exosome-mediated platinum drug resistance in breast cancer. Oxaplatin, carboplatin, cisplatin, lobaplatin, and nedaplatin are different platinum drugs used to treat different cancers. Tumor cell exosomes upregulate p-glycoprotein, which pumps the drug out of the cell; miR-423-5p alters the TME; exosomes released by Snail-induced EMT inhibit the activity of PTEN and BRCC3, thus downregulating the activity of NLRP3; the exosomes derived from SOD1 fibroblasts release miR-3960; this miRNA targets the BRSK2-induced phosphorylation of PIMREG by S16, thus inhibiting or decreasing the activity of the NF-Kβ signaling pathway—all these mechanisms by exosomes lead to cisplatin resistance. Note: Red bar denotes inhibition, downward red arrow denotes down-regulation, upward green arrow denotes upregulation.

**Figure 6 cancers-17-01167-f006:**
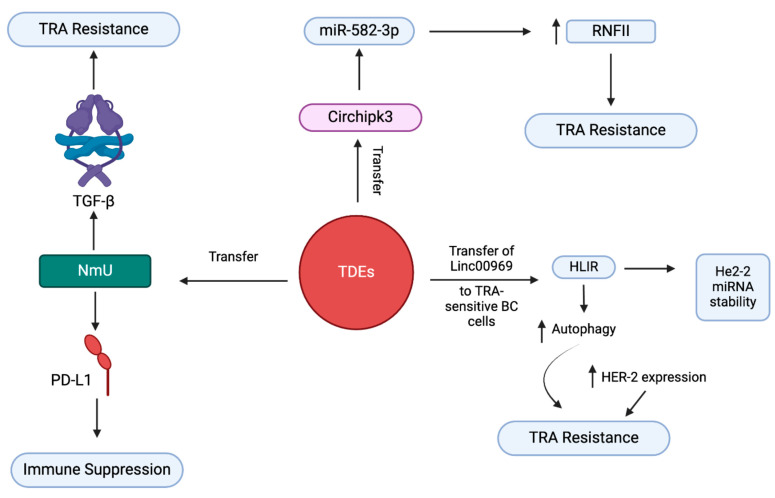
Exosome-mediated trastuzumab resistance in breast cancer. Exosomal Linc00969 induces HER-2 gene upregulation and promotes TRA resistance by triggering autophagy. Oncogenic circHIPK3 is regulated directly by miR-582-3p, which regulates the expression of RNF11, which induces drug resistance by selective degradation of proapoptotic proteins. Exosomes containing neuropeptide neuromedin U (NmU) enhance the release of TGF-β and PD-L1. TGF-β induces TRA resistance, and PD-L1 induces immune suppression in BC cells.

**Table 1 cancers-17-01167-t001:** Summary of exosomal contents/major pathways involved/mechanisms driving drug resistance in BC subtypes.

Exosomal Contents	Mechanism of Action	Consequences of Potential Therapeutic Targeting of Exosomes	Subtype	Ref
Circ_UBE2D2	Upregulates ER-α expression. Interacts with miR-2001-3p.	Inhibits the metastasis of BC cells.	Luminal A/B	[63]
miR-9-5p	Negatively regulates adiponectin.	Regulates the metastasis of BC cells.	Luminal A/B	[64]
miR-205	Inhibits caspase-3 by binding to E2F1. Inhibits AKT phosphorylation.	Enhances drug-induced apoptosis.	Luminal A/B	[68]
miR-181b-5p	Suppresses BCLAF1 expression and lowers p53/p21 levels.	Enhances the DOX-induced G1 cell cycle arrest and senescence.	TNBC	[72]
miR-155	Activates EMT.	Sensitizes the cells to PTX treatment.	TNBC	[74]
miR-222/miR-1246	Reduces p27kip1 synthesis and targets CCNG2.	Sensitizes the cells for drugs for treating ER+ BC.	Luminal A/B	[75]
miR-423-5p	miR-423-5p-dependent manner.	Regulates the alteration of the sensitivity of wild-type tumor cells for drugs and reduces drug resistance.	TNBC	[94]
miR-21	Downregulates NLRP3 inflammasome activity.	Sensitizes the cells for treating BC.	TNBC	[95]
miR-3960	Targets BRSK2 expression. Initiates NF-κB signaling pathway.	Inhibits the cancer progression and metastasis.	TNBC	[96]
Lnc00969	Interacts with HUR.	Sensitizes the cells for treating BC using TRA.	HER2+	[98]
circHIPK	Regulates RNF11 expression in association with miR-582-3p.	Sensitizes the cells for treating BC using DOX and TRA.	HER2+	[99]
miR-335-5p/-3p	Represses the genes involved in ERα signaling.	Sensitizes the drug resistance for drugs like TMX.	Luminal A/B	[101]
miR-128	Targets Bax gene.	Inhibits the cell proliferation.	Luminal A/B	[102]
miR-27a	Activates the Wnt/β-catenin signaling pathway.	Controls the proliferation, migration, and invasion of BC cells.	HER2+	[102]
UCA1 (lncRNA)	Activates mTOR signaling pathway.	Reduces the drug resistance.	Luminal A/B	[103]
miR-27b	AKT/NF-κB signaling cascade.	Controls the tumor growth.	HER2+	[104]
miR-21	Regulates PDCD4 protein levels.	Inhibits the metastasis to the bone.	TNBC	[105]

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
