# Peer review of "Exosome-Mediated Cellular Communication in the Tumor Microenvironment Imparts Drug Resistance in Breast Cancer"

_cancers, 2025, doi:10.3390/cancers17071167_

Round 1
Reviewer 1 Report
Comments and Suggestions for Authors
The review by Malla et al. provides a survey of the literature about the potential role of exosomes in the resistance of breast cancers to drug treatments. The major weaknesses in the review are:
- The Discussion section [lines 578-620] points out that there are several reviews of this area already and it is not really clear what this new manuscript adds. This manuscript is far too much of an uncritical survey of what has been published in the field and does not try to analyze the strength and weaknesses of what has been reported to give the reader insight into which of the many potential mediators are likely to be significant as compared to those that are likely incidental findings and reports. The lack of critique of the published literature means that this manuscript is really a survey, not a review.
- In a couple of places, the authors mention that mechanism of drug resistance is through up-regulation of the drug target [ER-alpha, line232; HER-2, line 452]. Since sensitivity to targeted treatments is typically predicted through robust expression of the target, this represents missed opportunities for the authors to analyze rather than just describe previous findings.
- The authors include tamoxifen and trastuzumab as chemotherapeutic agents when they are conventionally described as targeted. This removes the possibility of discussing whether the drug resistance mechanisms for targeted agents have a different theme from those for chemotherapeutic/cytotoxic agents.
- The definition of the TME is incomplete [lines 93-101] because it focuses only on the cellular components. Non-cellular components such as pH and pO2 are known to have very significant effects on drug resistance and should be at least acknowledged. For example, there is significant literature on the effects of those parameters on exosome production.
Additional problems with manuscript include:
- It is incorrect to state that breast is the most commonly diagnosed cancer. What about skin cancer? [line 36]
- The sentence on heterogeneity [line 48] does not make sense.
- Is it true that hormone-receptor breast cancer is less responsive to chemotherapy [line 57]? No citation is provided to support that assertion.
- The authors mention of their own previous work on gut microbiota [lines 106-109] is awkward and does not fit into a paragraph about tumor associated macrophages. Inclusion of this area in this review would need justification of its significance for breast cancer.
- Legend to fig. 5 describes red bar, red arrow, green arrow [lines 414-415]. None of those are clear in the figure.
- The description of effects on immune checkpoint inhibitors is confusing [e.g., lines 501-503]. The authors should likely distinguish between potential effects on Treg vs other T cells.
- The acronyms in Table 1, particularly for the drug names, should be defined within the Table and not require the reader to hunt through the text to interpret it.
Typos/confusing wordings include:
- "care' " [line 61]
- "due either" [line 70]
- "bringing deactivation" [line 79]
- "-resistant" [line 286]
- sentence about PTX action [lines 326-328] is very confusing
- wording about how PTX "makes the cancer cells resistant" is confusing [line 356]
- Using "proved" for a particular study [line 358] is a high bar that implies that it was challenged and particularly controversial.
- sentence about survival of recipient cells [lines 384-386] is confusing
- why is sentence lines 389-390 in bold?
The English is mostly ok. The sentences are not too long. Some specific confusing areas are listed above.
Author Response
Reviewer 1.
Comments and Suggestions for Authors
The review by Malla et al. provides a survey of the literature about the potential role of exosomes in the resistance of breast cancers to drug treatments. The major weaknesses in the review are:
Comment 1. The Discussion section [lines 578-620] points out that there are several reviews of this area already and it is not really clear what this new manuscript adds. This manuscript is far too much of an uncritical survey of what has been published in the field and does not try to analyze the strength and weaknesses of what has been reported to give the reader insight into which of the many potential mediators are likely to be significant as compared to those that are likely incidental findings and reports. The lack of critique of the published literature means that this manuscript is really a survey, not a review.
Response. As per the suggestion of the reviewer, discussion section was revised, and changes are highlighted.
Comment 2. In a couple of places, the authors mention that mechanism of drug resistance is through up-regulation of the drug target [ER-alpha, line232; HER-2, line 452]. Since sensitivity to targeted treatments is typically predicted through robust expression of the target, this represents missed opportunities for the authors to analyze rather than just describe previous findings.
Response. As per the suggestion of the reviewer, previous findings are analyzed the same was incorporated in the revised manuscript and changes were highlighted.
Comment 3. The authors include tamoxifen and trastuzumab as chemotherapeutic agents when they are conventionally described as targeted. This removes the possibility of discussing whether the drug resistance mechanisms for targeted agents have a different theme from those for chemotherapeutic/cytotoxic agents.
Response. As per the suggestion of the reviewer, tamoxifen and trastuzumab are corrected as targeted therapeutics in the page No.8 and line No 264 & Page No.15 and line No. 498 of the revised manuscript and changes are highlighted.
Comment 4. The definition of the TME is incomplete [lines 93-101] because it focuses only on the cellular components. Non-cellular components such as pH and pO2 are known to have very significant effects on drug resistance and should be at least acknowledged. For example, there is significant literature on the effects of those parameters on exosome production.
Response. As per the suggestion of the reviewer, the manuscript was revised by incorporating the effect of non-cellular components such as ECM, low pH and hypoxia on exosome release and uptake, and changes are highlighted in the page No.3, para 2 and line 104-122.
Additional problems with manuscript include:
Comment 5. It is incorrect to state that breast is the most commonly diagnosed cancer. What about skin cancer? [line 36]
Response. Skin cancer is most common cancer overall, while Breast cancer is most commonly diagnosed women cancer. As per the suggestion of the reviewer, the manuscript was revised, and changes are highlighted in page 2, para 1, line 46.
Comment 6. The sentence on heterogeneity [line 48] does not make sense.
Response. As per the suggestion of the reviewer, the manuscript was revised, and changes are highlighted in page 2, para 2 and line 59.
Comment 7. Is it true that hormone-receptor breast cancer is less responsive to chemotherapy [line 57]? No citation is provided to support that assertion.
Response. It is incorrect. As per the suggestion of the reviewer, the manuscript was revised, and changes are highlighted.
Comment 8. The authors mention of their own previous work on gut microbiota [lines 106-109] is awkward and does not fit into a paragraph about tumor associated macrophages. Inclusion of this area in this review would need justification of its significance for breast cancer.
Response. As per the suggestion of the reviewer, the manuscript was revised, and changes are highlighted in page 3, para 3 and lines 128-132.
Comment 9. Legend to fig. 5 describes red bar, red arrow, green arrow [lines 414-415]. None of those are clear in the figure.
Response. As per the suggestion of the reviewer, the figure 5 was revised.
Comment 10. The description of effects on immune checkpoint inhibitors is confusing [e.g., lines 501-503]. The authors should likely distinguish between potential effects on Treg vs other T cells.
Response. As per the suggestion of the reviewer, the manuscript was revised, and changes are highlighted in the page No. 18, para 4 and lines 605-610.
Comment 11. The acronyms in Table 1 (now it changed to table 2), particularly for the drug names, should be defined within the Table and not require the reader to hunt through the text to interpret it.
Response. As per the suggestion of the reviewer, the table was revised, and changes are highlighted.
Comment 12. Typos/confusing wordings include:
- "care' " [line 61]
- "due either" [line 70]
- "bringing deactivation" [line 79]
- "-resistant" [line 286]
- sentence about PTX action [lines 326-328] is very confusing
- wording about how PTX "makes the cancer cells resistant" is confusing [line 356]
- Using "proved" for a particular study [line 358] is a high bar that implies that it was challenged and particularly controversial.
- sentence about survival of recipient cells [lines 384-386] is confusing
- why is sentence lines 389-390 in bold?
Response. As per the suggestion of the reviewer, the changes are made in the revised manuscript and changes are highlighted.
Reviewer 2 Report
Comments and Suggestions for Authors
The review test is devoted to the study of the role of exosomes in the development of breast cancer resistance to known chemotherapeutic agents. The work examines the molecular mechanisms of the effect of exosomes on cells that make up the tumor microenvironment, the functional changes of which are important for tumor progression and metastasis. Possible ways to reduce or block the influence of exosomes on events occurring in the tumor are also given.
In my opinion, the text needs significant revision.
The text repeatedly talks about the development of drug resistance, while it mainly discusses resistance to treatment, which consists of inactivation of the drug molecule (reducing its concentration in the cytoplasm), activation of proliferation (maintaining the viability of tumor cells), increased metastasis, or suppression of the immune system response. However, all the facts mentioned in the text are presented mixed up and often without specifying which cells secrete exosomes that have the said effect, which cells are affected, and by what specific mechanism resistance to treatment develops in this case. It is also not specified whether this occurs under the influence of the drug or at any time during tumor development.
Similar questions arise throughout the text. Here are just a few of them.
Section 2.1 provides general information about the formation of exosomes by any cells. Are there any features of exosome formation common specifically to tumor cells? Tumor microenvironment cells?
Do the mechanisms of exosome action described in Section 2.2 develop only in response to the presence of drugs or at any time? And which cells secrete exosomes in each specific case?
Line 114 should provide references from 24-25 and there should be several of them. References should also be provided on line 123.
Line 176 How are drug resistance and TC cells related?
Line 226 A reference should be provided. An explanation of the mechanism of this phenomenon is required. Do the cell's own exosomes reduce the concentration of the drug or do exosomes from other cells?
Line 228 and others The mechanism of action of each specific drug is important for understanding the effect that exosomes can have on it.
In Figure 1, some arrows do not point to the correct area (for example, EPHA2). Fibroblasts usually have a characteristic spindle shape, which is depicted incorrectly in the figure.
In Figure 2, the word Adiponectin is misspelled. The arrow leading from apoptosis to drug resistance raises questions. That is, cell apoptosis leads to the development of resistance?
Overall, the text of the paper is similar to reference [30] in many ways, but is much less structured and understandable.
Author Response
Reviewer 2.
Comments and Suggestions for Authors
The review test is devoted to the study of the role of exosomes in the development of breast cancer resistance to known chemotherapeutic agents. The work examines the molecular mechanisms of the effect of exosomes on cells that make up the tumor microenvironment, the functional changes of which are important for tumor progression and metastasis. Possible ways to reduce or block the influence of exosomes on events occurring in the tumor are also given.
In my opinion, the text needs significant revision.
Comment 1. The text repeatedly talks about the development of drug resistance, while it mainly discusses resistance to treatment, which consists of inactivation of the drug molecule (reducing its concentration in the cytoplasm), activation of proliferation (maintaining the viability of tumor cells), increased metastasis, or suppression of the immune system response. However, all the facts mentioned in the text are presented mixed up and often without specifying which cells secrete exosomes that have the said effect, which cells are affected, and by what specific mechanism resistance to treatment develops in this case. It is also not specified whether this occurs under the influence of the drug or at any time during tumor development.
Response. As per the suggestion of the reviewer, the cells secreting the exosomes was incorporated in the revised manuscript, and changes are highlighted.
Comment 2. Similar questions arise throughout the text. Here are just a few of them.
Section 2.1 provides general information about the formation of exosomes by any cells. Are there any features of exosome formation common specifically to tumor cells? Tumor microenvironment cells?
Response. As per the suggestion of the reviewer, the manuscript was revised, and changes are highlighted in the page No. 4, para 2, line No. 147-151.
Comment 3. Do the mechanisms of exosome action described in Section 2.2 develop only in response to the presence of drugs or at any time? And which cells secrete exosomes in each specific case?
Response. These mechanisms exist constitutively but drug presence intensifies them. The same was incorporated in the revised manuscript, and changes are highlighted in the Page No.4, para 2 and line 142-157.
Comment 4. Line 114 should provide references from 24-25 and there should be several of them.
Response. As per the suggestion of the reviewer, references are included in the revised manuscript and changes are highlighted in page No.4, line 142.
Comment 5. References should also be provided on line 123.
Response. As per the suggestion of the reviewer, references are included in the revised manuscript and changes are highlighted in page No.4, line 155
.
Comment 6. Line 176 How are drug resistance and TC cells related?
Response. As per the suggestion of the reviewer, the relation of drug resistance and TC cells was described in the revised manuscript and the same was highlighted in page 6 and line 211-217.
Comment 7. Line 226 A reference should be provided. An explanation of the mechanism of this phenomenon is required. Do the cell's own exosomes reduce the concentration of the drug or do exosomes from other cells?
Response. As per the suggestion of the reviewer, the reference and mechanism of p-glycoprotein mediated drug resistance was included in the revised manuscript, and changes are highlighted in page No.7 and line 267.
Comment 8. Line 228 and others. The mechanism of action of each specific drug is important for understanding the effect that exosomes can have on it.
Response. As per the suggestion of the reviewer, mechanism of action of each specific drug was included in the revised manuscriptand changes are highlighted in the line 270 of page No.7 ; 316 line of page No 9; line 456 of page No 12 and line 511 of page 14.
Comment 9. In Figure 1, some arrows do not point to the correct area (for example, EPHA2). Fibroblasts usually have a characteristic spindle shape, which is depicted incorrectly in the figure.
Response. As per the suggestion of the reviewer, figure 1 was modified.
Comment 10. In Figure 2, the word Adiponectin is misspelled. The arrow leading from apoptosis to drug resistance raises questions. That is, cell apoptosis leads to the development of resistance?
Response. As per the suggestion of the reviewer, the figure 2 was modified.
Comment 11. Overall, the text of the paper is similar to reference [30] in many ways but is much less structured and understandable.
Response. As per the suggestion of the reviewer, the manuscript was revised, and changes are highlighted.
Reviewer 3 Report
Comments and Suggestions for Authors
This review article by Malla et al. focuses on the complex interaction of exosomes with the tumor microenvironment and the consequences of this communication for conferring drug resistance of breast cancer. This is an excellently written manuscript with well prepared accompanying figures to illustrate major mechanisms of exosome-dependent drug resistance in BC. With it, the authors provide an extensive and well researched overview on this timely topic that should be of great interest to the readers of Cancers.
My only criticism is related to the different subtypes of BC. Based on the information given in the text and figures, it is not entirely clear inasmuch exosomal contents/major pathways involved/mechanisms driving drug resistance may differ between BC subtypes and what consequences this may have for the potential therapeutic targeting of exosomes in BC. Based on available data and further hypotheses, the authors should try to elaborate on this aspect as much as possible to further improve the manuscript.
Fig. 5 The size of the figure could be aligned to the other figures (font sizes). The word "suppression" should be added to immune in the lower right box.
Author Response
Reviewer 3.
Comments and Suggestions for Authors
This review article by Malla et al. focuses on the complex interaction of exosomes with the tumor microenvironment and the consequences of this communication for conferring drug resistance of breast cancer. This is an excellently written manuscript with well-prepared accompanying figures to illustrate major mechanisms of exosome-dependent drug resistance in BC. With it, the authors provide an extensive and well researched overview on this timely topic that should be of great interest to the readers of Cancers.
Comment 1. My only criticism is related to the different subtypes of BC. Based on the information given in the text and figures, it is not entirely clear in as much exosomal contents/major pathways involved/mechanisms driving drug resistance may differ between BC subtypes and what consequences this may have for the potential therapeutic targeting of exosomes in BC. Based on available data and further hypotheses, the authors should try to elaborate on this aspect as much as possible to further improve the manuscript.
Response. As per the suggestion of the reviewer, exosomal contents/major pathways involved/mechanisms driving drug resistance may differ between BC subtypes and what consequences this may have for the potential therapeutic targeting of exosomes in BC was included in the table 1 and elaborated in the revised manuscript and and changes are highlighted.
Comment 2. Fig. 5 The size of the figure could be aligned to the other figures (font sizes). The word "suppression" should be added to immune in the lower right box.
Response. As per the suggestion of the reviewer, corrections are done in the figure 5.
Round 2
Reviewer 2 Report
Comments and Suggestions for Authors
Dear authors, thank you for the changes and clarifications, they have filled in many blank spots that prevented adequate perception of the text of the work.
I still have a few small comments.
Figure captions should contain an explanation of all the symbols and abbreviations used in the figure, and not retell the results.
What does the expression "drug-resistant cargo" used on line 259 mean?
The text contains a wealth of information about exosomes, which are secreted by cells that make up a tumor. The functions of exosomes are considered in terms of their involvement in the formation of resistance to chemotherapeutic agents. Generalization of these data opens up opportunities for finding new methods of combating tumor resistance to known treatment methods. The first part contains information about the role of exosomes in transforming the tumor microenvironment, the second describes specific mechanisms of interaction between exosomes and drugs. The presence of two clearly separated parts is confusing. What is the main topic of the review? The ways in which tumors "survive" despite all methods of their destruction? Or exosomes?
Author Response
Comment 1. Figure captions should contain an explanation of all the symbols and abbreviations used in the figure, and not retell the results.
Response: As per the suggestion of the reviewer, we changed all the figure captions which are highlighted red.
Comment 2. What does the expression "drug-resistant cargo" used on line 259 mean?
Response: As per the suggestion of the reviewer, we changed "drug-resistant cargo" to "molecules imparting drug resistance" highlighted red in page 259.
Comment 3. The text contains a wealth of information about exosomes, which are secreted by cells that make up a tumor. The functions of exosomes are considered in terms of their involvement in the formation of resistance to chemotherapeutic agents. Generalization of these data opens up opportunities for finding new methods of combating tumor resistance to known treatment methods. The first part contains information about the role of exosomes in transforming the tumor microenvironment, the second describes specific mechanisms of interaction between exosomes and drugs. The presence of two clearly separated parts is confusing. What is the main topic of the review? The ways in which tumors "survive" despite all methods of their destruction? Or exosomes?
Response:
We appreciate the reviewer's insightful comment regarding the structure of the review. We acknowledge that the initial organization, separating exosome-mediated TME remodeling and exosome-drug interactions.
To clarify, the main focus of this review is to elucidate the multifaceted role of exosomes in promoting drug resistance in breast cancer. While the first section provides a foundational understanding of exosome-mediated communication within the TME, it serves primarily to contextualize how these altered interactions ultimately lead to enhanced resistance against a variety of therapies. Therefore, the second part is the main focus and has also been strengthened more with new papers with more new directions for researchers. We believe that both sections are fundamentally intertwined and centered on the key role exosomes confer.